# Effect of Human Platelet Lysate as Cultivation Nutrient Supplement on Human Natal Dental Pulp Stem Cell In Vitro Expansion

**DOI:** 10.3390/biom12081091

**Published:** 2022-08-08

**Authors:** Nela Pilbauerova, Jan Schmidt, Tereza Suchankova Kleplova, Tomas Soukup, Jakub Suchanek

**Affiliations:** 1Department of Dentistry, Charles University, Faculty of Medicine in Hradec Kralove, University Hospital Hradec Kralove, 500 05 Hradec Kralove, Czech Republic; 2Department of Histology and Embryology, Charles University, Faculty of Medicine in Hradec Kralove, 500 03 Hradec Kralove, Czech Republic

**Keywords:** stem cell cultivation, human platelet lysate, mesenchymal stem cells, human natal stem cells, fetal bovine serum, culture medium nutrient supplement, regenerative medicine

## Abstract

Despite several scientific or ethical issues, fetal bovine serum (FBS) remains the standard nutrient supplement in the mesenchymal stem cell cultivation medium. Cell amplification plays an important role in human stem cell therapies. Increasing interest in this field has supported attempts to find suitable human alternatives to FBS for in vitro cell propagation. Human platelet lysate (hPL) has recently been determined as one of them. Our study aimed to evaluate the influence of 2% hPL in the growth medium for in vitro expansion of human natal dental pulp stem cells (hNDP-SCs). The effect was determined on proliferation rate, viability, phenotype profile, expression of several markers, relative telomere length change, and differentiation potential of four lineages of hNDP-SCs. As a control, hNDP-SCs were simultaneously cultivated in 2% FBS. hNDP-SCs cultivated in hPL showed a statistically significantly higher proliferation rate in initial passages. We did not observe a statistically significant effect on mesenchymal stem cell marker (CD29, CD44, CD73, CD90) or stromal-associated marker (CD13, CD166) expression. The cell viability, relative telomere length, or multipotency remained unaffected in hNDP-SCs cultivated in hPL-medium. In conclusion, hPL produced under controlled and standardized conditions is an efficient serum supplement for in vitro expansion of hNDP-SCs.

## 1. Introduction

The occurrence of natal and neonatal teeth in humans is an uncommon anomaly of premature tooth eruption. Natal teeth are teeth present at birth, and neonatal teeth are teeth that erupt within the first month of life [1]. They might further accompany various difficulties, such as pain on sucking and refusal to feed, faced by the mother or child [1,2]. Although the eruption of the lower deciduous incisors is normal at birth in many mammals, natal and neonatal teeth are rare in humans. The incidence of natal teeth ranges from 1:2000 to 1:3500 live birth [1,2,3,4,5]. Only 1–10% of natal and neonatal teeth are supernumerary, and more than 90% of natal teeth and neonatal teeth are prematurely erupted primary teeth [6]. Natal teeth are three times more common than neonatal teeth. They most frequently occur in the mandibular central incisor region.

The pulp of natal teeth is a source of a unique type of dental pulp-related stem cells that display some of the characteristics of pluripotency [7,8]. In comparison with dental pulp stem cells isolated from the permanent (DPSCs) of deciduous teeth (SHED), human natal dental pulp stem cells (hNDP-SCs) have a higher proliferation rate and express similar surface markers CD13, CD29, CD44, CD73, CD90, CD146, and CD166. Interestingly, hNDP-SCs express detectable levels of factors, such as Nanog, octamer-binding transcription factor 4 (Oct-4), transcription factors Sox-2, and forkhead box protein FoxD3 [7,8]. They also show positivity for osteogenic, myogenic, chondrogenic, and neural markers under cultivation in the standard conditions [7]. That determines their immature nature and wide differentiation potential, which are considered the most valuable in regenerative or reparative cell therapies.

Most cell-based therapies require a high cell number. Therefore, after their isolation, stem cells must be amplified ex vivo under controlled conditions without any effects on their “stemness” characteristics. Most laboratory protocols for stem cell expansion use the cultivation medium supplemented with fetal bovine serum (FBS). FBS contains essential cell proliferation and maintenance components, such as hormones, vitamins, transport proteins, trace elements, spreading, and growth factors [9]. Unfortunately, the use of FBS is associated with several problematic issues. In light of scientific, consistency, and ethical issues, the search for human alternatives to FBS has become a major goal in recent years [10,11,12,13]. However, chemically-defined serum-free media have not yet been developed for every cell line or primary culture at this time due to issues including heterogeneity among cell types and potential effects on cell phenotype or reproducibility. A human platelet lysate (hPL) has been identified as an autologous replacement for animal-derived supplements such as FBS in the expansion of stem and progenitor cells for tissue engineering applications and cell therapies [14,15]. hPL is prepared from platelet-rich plasma (PRP), either derived from pooled buffy coat–derived platelet concentrates of whole blood or from apheresis. In contrast to PRP, which requires a more complicated manufacturing process, hPL can be generated from common platelet units by a simple freeze-thawing procedure. To avoid extensive aggregate formation and to deplete potential antigens, the platelet fragments are removed by centrifugation [16,17]. Because hPL is a platelet element free, the immunological reaction connected with allogeneic products can be obviated. On the other hand, it contains a high level of several growth factors, including insulin-like growth factor 1 (IGF-1), transforming growth factor-beta (TGF-β1, TGF-β2), platelet-derived growth factor (PDGF), fibroblast growth factors (FGF), vascular endothelial growth factor (VEGF) and epidermal growth factor (EGF) [18,19]. Some of these signal regulators have also been identified to regulate osteogenesis of dental pulp stem cells or stem cells from the gingiva or periodontal ligaments [20,21,22,23]. Therefore, hPL has been shown to promote cell growth, which was mostly superior to standard cultures supplemented with FBS. In contrast, the differentiation capacity of progenitor cells seems not to be affected [24,25,26]. Unfortunately, it must be taken into consideration that hPL composition varies in the amount of plasma and range of growth factors [26]. Therefore, the research studies investigating hPL in stem cell cultivation bring conflicting results. Variation between individual platelets can be reduced by pooling platelet units for the use of hPL under good manufacturing practice conditions.

The study aimed to evaluate the influence of hPL as a nutrient supplement on the proliferation rate, viability, phenotype profile, several progenitor cell marker expression, relative telomere length difference, and differentiation potential of hNDP-SCs cultivated in vitro.

## 2. Materials and Methods

Study guidelines were approved by the Ethics Committee of the University Hospital Hradec Kralove (201812 S07P, approved 8 November 2018). Legal representatives of donors signed informed written consents before natal tooth collection.

### 2.1. Natal Teeth Collection and hNDP-SCs Isolation

Four lineages of hNDP-SCs were isolated from natal teeth obtained from healthy newborns—female (11 days old) and male (4 days old) born at the Department of Obstetrics and Gynecology, University Hospital Hradec Kralove, Czech Republic. The natal teeth were immediately after their collection immersed in a 0.2% chlorhexidine gluconate solution for 30 s for decontamination. Subsequently, teeth were transported in a tube containing a transport medium consisting of water for injection (Bieffe Medital, Grosotto, Italy), 10% Hank’s balanced salt solution (HBSS) (Invitrogen, Waltham, MA, USA), 200 μg/mL gentamicin (Invitrogen), 200 U/mL penicillin (Invitrogen), 200 μg/mL streptomycin (Invitrogen), and 1.25 μg/mL amphotericin (Sigma-Aldrich, St. Louis, MO, USA). During the transportation, the teeth were fully immersed in a medium, and the temperature was kept at 4 °C. The hNDP-SC isolation was performed at a sterile laminar box at the Department of Histology and Embryology, Charles University, Faculty of Medicine in Hradec Kralove, on the same day the teeth were harvested. Due to absent roots, the pulp tissues were simply removed from pulp cavities using a sterile probe and tweezers. Then, they were enzymatically digested as previously described [27]. Briefly, pulp tissues were minced by scissors and homogenized in a mini tissue grinder (Radnoti, Covina, CA, USA) with an isotonic solution (phosphate-buffered saline, PBS, (Sigma-Aldrich). The tissue homogenate was transferred into a tube and enzymatically digested using 0.05% trypsin (Gibco, Thermo Fisher Scientific, Foster City, CA, USA) for 10 min at 37 °C. After centrifugation (2000 rpm/5 min, 600 g), the pellet was resuspended in an adherent tissue culture dish (TPP, Sigma-Aldrich).

### 2.2. Culture Media

Table 1 illustrates the composition of each culture medium.

Cells seeded in adherent surface culture dishes were cultivated under standard conditions in a humidified atmosphere containing 5% CO2 at 37 °C. Every three days, we removed the detritus and non-adherent elements by washing dishes with a phosphate-buffered saline (PBS, (Sigma-Aldrich) and exchanged each culture medium. Each lineage was detached from the adherent surface using 0.05% Trypsin-EDTA (Gibco, Thermo Fisher Scientific) and passaged when hNDP-SCs reached approximately 70% confluence. Subsequently, it was reseeded in a final concentration of 5000 cells/cm^2^. All cell lines were terminated in the 14th passage (14p).

The hPL was obtained from the Transfusion department, University Hospital Hradec Kralove, Czech Republic. Briefly, the preparation protocol started with platelet-rich plasma (PRP) units derived from buffy coats. After a sterility check, PRP units were frozen to at least −20 °C in the original storage bags. When the bacterial test was negative, hPL units were thawed at 37 °C (water bath) until the ice clots disappeared. To decrease individual platelet variation, one hPL unit was pooled from five randomly chosen units donated by healthy individuals, subject to guidelines for the selection of blood donors. To increase the rate of platelet fragmentation and the number of released growth factors, hPL units were further re-frozen and re-thawed. Subsequently, the hPL units were centrifuged to discard the platelet pellets and keep the supernatant rich in factors.

### 2.3. Effect on hNDP-SC Proliferation and Viability

We count a total cell count in each passage using a Z2-Counter (Beckman Coulter, Miami, FL, USA). For each measurement, we analyzed 100 µL/1 mL cell suspension (approximately 1.5 × 10^6^ cells/1 mL). Proliferation capacity was evaluated as population doublings (PDs) and population doubling time in hours (Equation (1)). Proliferation activity was compared from the 1st passage when hNDP-SCs were seeded into two different growth media.
PD = log2 (N_x_/N_1_)(1)

Equation (1). N_x_ is the total passage cell count calculated using the Z2-Counter (Beckman Coulter), and N_1_ is the initial cell count seeded into the culture dish (5000 cells/cm^2^).

To determine the number of viable hNDP-SCs cells from each sample in the 3rd and 11th passage, we used the trypan dye exclusion method by Vi-Cell analyzer (Beckman Coulter). This method is based on the principle that viable cells do not take up the trypan blue dye, whereas non-viable cells do due to disturbed cell membrane. For each analysis, we utilized 250 µL of cell suspension (approximately 1.5 × 10^6^ cells/1 mL) and 250 µL PBS.

### 2.4. Effect on hNDP-SC Phenotype Profile and Specific Factor Expressions

To identify that hNDP-SCs maintain their phenotypic characteristics after growth in different media, undifferentiated hNDP-SCs were subjected to flow cytometric analysis using a Cell Lab Quanta analysis (Beckman Coulter). hNDP-SCs from 5th passage (10^4^ cells per one CD marker analyses) were stained with primary immunofluorescence antibodies conjugated with phycoerythrin (PE) or fluorescein (FITC) against the following markers: CD10 (Neprilysin: membrane metallo-endopeptidase; CB-CALLA, eBioscience, San Diego, CA, USA), CD13 (Alanyl aminopeptidase; WM-15, eBioscience), CD29 (Integrin beta-1; TS2/16, BioLegend, San Diego, CA, USA), CD34 (Transmembrane phosphoglycoprotein; 581 (Class 287 III), Invitrogen), CD44 (Cell-surface glycoprotein; MEM 85, Invitrogen), CD45 (Protein tyrosine phosphatase; HI30, Invitrogen), CD71 (Transferrin receptor protein 1; BioLegend, MEM-75), CD73 (5′-nucleotidase; AD2, BD Biosciences Pharmingen, Erembodegen, Belgium), CD90 (Thy-1; F15-42-1-5, Beckman Coulter), CD105 (Endoglin; SN6, 289, Invitrogen), CD146 (Melanoma cell adhesion molecule; TEA1/34, Beckman Coulter), CD166 (Activated leukocyte cell adhesion molecule; 3A6, Beckman Coulter), CD271 (Low-affinity nerve growth factor receptor; ME20.4, BioLegend). The percentage of positive cells was determined as a percentage of cells with higher fluorescence intensity than the upper 0.5% isotype immunoglobulin control.

To detect specific biomolecules within hNDP-SCs, we performed immunofluorescent staining. For this analysis, hNDP-SCs from the 7th passage were seeded in a concentration of 5000 cells per cm^2^ and cultivated in chamber slides (Nalge Nunc International Corporation, Rochester, NY, USA) for two days. Before immunostaining, the adherent cells were fixed using 10% formaldehyde. After thorough washing with phosphate-buffered saline (PBS), a 0.5% solution of Triton (250 mL Triton (Sigma-Aldrich) and 500 mL PBS) was used for 10 min to facilitate antibody penetration. Subsequently, samples were stained with primary antibodies against Beta3-tubulin (TU-20, 1:50, Exbio, Prague, Czech Republic), Nestin (10C2, 1:200, Chemi-Con, Nuremberg, Germany), neurofilaments (DA2, FNPT, prediluted, Zymed, South San Francisco, CA, USA), and Nanog (rabbit polyclonal, 1:200, Abcam, Cambridge, UK). After washing, pellets were incubated with appropriate secondary antibodies conjugated with fluorochromes for 30 min to visualize the antigen-binding sites. Cell nuclei were counterstained with 4′-6-diamidino-2-phenylindole (DAPI, Sigma-Aldrich) for 5 min and observed samples with a BX51 Olympus microscope. Images were overlapped using Adobe^®^ Photoshop CC 2021 (Adobe Systems, San Jose, CA, USA).

### 2.5. Effect on Relative Telomere Length

To assess the effect of cultivation with hPL on the relative telomere length measured using real-time polymerase chain reaction (qPCR) in the 3rd and 14th passages (Equation (2)). The analysis protocol was used in previous studies [27,28,29]. After the DNA isolation using a DNeasy Tissue Kit (Qiagen, Hilden, Germany), we calculated its concentration in each sample using a spectrophotometer Nanodrop 1000 (Thermo Fisher Scientific, Waltham, MA, USA).
T/S = 2^−^^ΔCt^(2)

Equation (2). The formula for the relative telomere length calculation, where ΔCt = CT_telomere_ − CT_“single copy” gene_. The single gene (housekeeping gene) was a coding acidic ribosomal phosphoprotein 36B4.

We performed the qPCR in 96-well plates and analyzed each sample in triplicates at the same well position on an ABI 7500 HT detection system (Applied Biosystems, Foster City, CA, USA). Each 20 μL re-action consisted of 20 ng DNA, 1 × SYBR Green master mix (Applied Biosystems), 200 nM forward telomere primer (CGG TTT GTT TGG GTT TGG GTT TGG GTT TGG GTT), and 200 nM reverse telomere primer (GGC TG TCT CCT TCT CCT TCT CCT TCT CCT TCT CCT). We used the following primer pairs for the housekeeping gene analysis: 36B4u, CAG CAA GTG GGA AGG TGT AAT CC; 36B4d, CCC 135 ATT CTA TCA TCA ACG GGT ACA A. The cycling of each qPCR analysis (for both telomere and housekeeping gene) started with a ten-minute cycle at 95 °C, followed by 15-s cycles at 95 °C, ending with a one-minute cycle at 60 °C.

### 2.6. Effect on hNDP-SC Multipotency

For osteogenic, chondrogenic, or adipogenic induction, hNDP-SCs from the 4th passage were seeded into separated culture dishes and grown to 80–100% confluence. Subsequently, they were incubated in differentiation media.

#### 2.6.1. Osteogenic Differentiation In Vitro

After hNDP-SCs reached confluence, we induced osteogenic differentiation with the FBS-free differentiation medium containing α-MEM (Sigma-Aldrich), 0.5 mM ascorbic acid (Bieffe Medital), 10 mM of β-glycerophosphate (Sigma-Aldrich), 0.1 μM of dexamethasone (Bieffe Medital), and 10% hPL (Transfusion Department, University Hospital Hradec Kralove, Czech Republic). As a standard osteogenic differentiation medium, we used the Human Mesenchymal Stem Cell Osteogenic Differentiation Medium BulletKit^TM^ (Lonza, Basel, Switzerland). Both media were exchanged after the washing step using PBS twice a week. hNDP-SCs were cultivated under differentiation conditions for 3 weeks. At the end of the third week, the pellets were fixed using 10% formalin, dehydrated in ascending concentrations of ethanol, embedded in paraffin, and cut into 7 μm thick sections. Osteogenic differentiation was assessed via staining with Alizarin Red S and von Kossa staining to locate calcium deposits in the extracellular matrix. Osteocalcin was detected using immunocytochemistry. After deparaffination, samples were exposed to a primary mouse IgG antibody (1:50, Millipore, Burlington, MA, USA) and donkey anti-mouse secondary IgG antibody (1:250, Jackson ImmunoResearch Labs, West Grove, PA, USA).

#### 2.6.2. Chondrogenic Differentiation In Vitro

In hNDP-SCs cultivated in FBS free cultivation medium, we induced chondrogenic differentiation using a medium containing α-MEM (Sigma-Aldrich), 0.5 mM ascorbic acid (Bieffe Medital), 10 mM of β-glycerophosphate (Sigma-Aldrich), 0.1 μM of dexamethasone (Bieffe Medital), and 50 ng/mL TGF-β1 (Stem Cell Technologies, Vancouver, BC, Canada). Human Mesenchymal Stem Cell Chondrogenic Differentiation Medium BulletKit^TM^ (Lonza) was used as a control medium supplemented with FBS. We changed both media twice a week and cultivated cells for 3 weeks. After three weeks, we prepared 7 μm thick paraffin sections from fixed differentiated cell pellets. Afterward, we detected collagen and procollagen in the extracellular matrix using blue Masson’s trichrome stain and specified type II collagen with immunochemical detection. Slices were incubated with a primary mouse IgM antibody (1:500, Sigma-Aldrich) and Cy3TM-conjugated goat anti-mouse secondary IgM antibody. Cell nuclei were counterstained with 4′-6-diamidino-2-phenylindole (DAPI, Sigma-Aldrich). Furthermore, we also stained acid mucopolysaccharides in the chondrogenic matrix using Alcian blue histological staining.

#### 2.6.3. Adipogenic Differentiation In Vitro

To induce pro-adipogenic conditions, HNDP-SCs grown without FBS were cultivated using the Mesenchymal Stem Cell Adipogenic Differentiation Medium kit (Cyagen Biosciences, Santa Clara, CA, USA), differentiation Basal Medium A and B containing 10% hPL instead of FBS. hMSC Adipogenic medium kit (Lonza), induction medium, and maintenance medium induced adipogenesis in cells cultivated in a medium with FBS. Cells were exposed to pro-adipogenic conditions for four weeks. Two different media from each kit were subsequently used and switched every three days for three weeks. Last week, hNDP-SCs were cultivated only in the hMSC Adipogenic Maintenance medium or Differentiation Basal Medium B. Cultures were then fixed with 10% formalin and rinsed with 50% ethanol. The presence of intracellular lipid droplets, which indicates that adipogenic differentiation occurred, was confirmed by Oil Red O staining.

### 2.7. Statistical Analysis

The data are presented as the mean ± SD. All statistical analyses were performed using the statistical software GraphPad Prism 9.3.0 (San Diego, CA, USA). The Shapiro–Wilk or Kolmogorov–Smirnov tests were used for normal distribution evaluations. The statistical significances (* *p* < 0.05, ** *p* < 0.01, *** *p* < 0.001, **** *p* < 0.0001) were calculated using either the paired *t*-test for continuous variables or the Wilcoxon matched-pairs test for nonparametric variables.

## 3. Results

Independently on the culture medium, hNDP-SCs initially had a rounded spindle-like shape with elongated processes reaching the surrounding cells (Figure 1).

### 3.1. Effect on hNDP-SC Proliferation and Viability

After several passages, they became prolonged and more spindled. Interestingly, during initial passages (mainly 2p–5p), hNDP-SCs grown in the hPL-culture medium proliferated faster than their counterparts. The average PDT (2p–5p) per passage for hNDP-SCs grown in the hPL-culture medium was 22.65 ± 0.10 h (average PD per passage (2p–5p) was 3.97 ± 0.35). The figure for hNDP-SCs grown in the FBS-culture medium was 44.69 ± 1.36 h (PD = 3.98 ± 0.38). This difference was statistically significant (*p* < 0.0001). The same trend, but less significant, was seen till the 8th passage when the proliferation rate of both groups became nearly the same. The proliferation rate of hNDP-SCs cultured in the hPL-culture medium was faster again between the 10th and 12th passage, but with a lower figure for PD in the 12th passage (cumulative PD in the 12th passage was 53.94 ± 0.89) compared with the hNDP-SCs cultured in the FBS-culture medium (55.87 ± 0.95). At the end of cultivation, the proliferation rate for hPL-cultivated hNDP-SCs c slowed down. Figure 2a,b illustrates the proliferation capacity of both groups displaying cumulative PDs and PDT in hours.

The percentages of viable cells were established using the trypan blue exclusion method in the 3rd and 11th passages. The figures for hPL-cultivated and FBS-cultivated hNDP-SCs were over 90% for the entire cultivation time (Figure 3).

### 3.2. Effect on hNDP-SC Phenotype Profile and Specific Factor Expressions

Defined markers exist that specifically and uniquely identify mesenchymal stem cells. The flow cytometry analysis of all common mesenchymal, hematopoietic stem cell markers indicated that hNDP-SCs grown in the hPL-culture medium were highly positive (>95%) to most of the tested markers. On the other hand, the same trend was not observed in FBS-cultivated hNDP-SCs. We did not observe statistically significant variances in CD markers defined as mesenchymal stem cell markers CD29, CD44, CD73, CD90, or stromal-associated markers CD13, CD166. Both groups showed high average expression of these markers (<90%). The protein tyrosine phosphatase (CD45) was highly expressed in hPL-cultivated hNDP-SCs, but lowly expressed in FBS-cultivated hNDP-SCs (>10%). The markers CD34, CD105, and CD146, differed statistically significantly in comparison between groups (Figure 4).

Remarkably, hNDP-SCs express detectable levels of the embryonic stem cell (ESC) markers or neuronal markers. We performed immunocytochemistry to determine whether this spontaneous ability is independent of culture medium composition. Undifferentiated hNDP-SCs were stained with primary antibodies against Beta3-tubulin (an early neuronal marker [30]), Nestin (neural progenitor marker [30]), neurofilaments (a late neuronal marker [30]), and Nanog (a marker known for its functions in ESC pluripotency, maintenance, and self-renewal [31]). The following pictures showed that the markers mentioned above were positively expressed in most of the cells independently whether we used FBS- or hPL-supplemented medium for their cultivation (Figure 5, Figure 6, Figure 7 and Figure 8).

### 3.3. Effect on Relative Telomere Length

Many studies have reported that the telomere length shortening is a hallmark of cell senescence, and the maintenance of their length is essential for cell self-renewal ability and differentiation potential [32,33]. Therefore, we performed qPCR to explore whether there are variances in relative telomere length between hNDP-SCs cultivated in two different media. We studied the differences in relative telomere length changes between the 3rd and 14th passages. Independently of the culture medium, we observed the statistically significantly shorter relative telomere length in the 14th passage than in the 3rd passage. The shortening was more noticeable in the case of hNDP-SCs cultivated in the standard FBS-culture medium (Figure 9).

### 3.4. Effect on hNDP-SC Multipotency

The Mesenchymal and Tissue Stem Cell Committee of the International Society for Cellular Therapy proposes minimal criteria to define human mesenchymal stem cells, where hNDP-SCs belong. One criterion is that they must differentiate into osteoblasts, adipocytes, and chondroblasts in vitro. We proved that hNDP-SCs cultivated in an hPL- or FBS-culture medium kept the ability to differentiate in osteogenic, chondrogenic, and adipogenic cell lines. To confirm our results, we used histologic staining and immunocytochemistry. The following Figure 10, Figure 11, Figure 12, Figure 13 and Figure 14 show our results.

## 4. Discussion

An increasing interest in mesenchymal stem cells and their role in regenerative and reparative medicine has brought many concerns and limitations that should be considered before their future broader use, especially in human medicine. One of the limitations is the widely used standard medium supplement and source of growth factors for cell culture, fetal bovine serum. Regenerative cells occur in low doses in origin tissues. Therefore, they must be amplified after isolation to obtain a suitable dose for their clinical application. hNDP-SCs are not an exception. Even though the risks of xenoimmunization against bovine antigens, the transmission of pathogens, and ethical issues associated with FBS collection are well known [9,10,11,34], FBS remains the standard growth factor supplement in most laboratory cultivation protocols. Therefore, it is a rational and multiple basic scientific research supported to find suitable human alternatives for in vitro cell propagation. This has become more important with the rapidly growing field of advanced cell therapy, where the use of FBS should be avoided regarding the international guidelines.

Over the past two decades, various human alternatives have been tested for their ability to sustain the proliferation and differentiation of cells in culture. Using human serum (HS) might seem the most straightforward solution. Unfortunately, it has been published that this method is unreliable, MSC proliferation is slow, and cells have difficulty reaching the desired confluence [35]. The platelet-rich plasma has been shown to enhance MSC proliferation [12,36,37,38], but the present debris in PRP might disturb cell culture. We concluded the same results in our previous study [39]. Although we came up with sufficient results in the hNDP-SC cultivation in 2% PRP, we had to deal with handling difficulties thanks to debris during the entire cultivation. Furthermore, it is necessary to activate thrombocytes to release growth factors. The use of human platelet lysate was first described in 2005 [16]. Thus far, hPL enriched in growth factors (such as platelet-derived growth factor) is particularly used for human MSCs, endothelial, and fibroblast cultures. Variations exist between individual hPL, but that can be solved by pooling. It is rarely distributed commercially [16,26]. Furthermore, all human blood-derived substituents remain a threat to the transmission of human diseases by known or unknown viruses such as human immunodeficiency virus and human T-lymphotropic virus. Nevertheless, these threats could be decreased by strict adherence to blood bank quality standards.

Our study aimed to verify the effect of hPL supplemented culture medium on hNDP-SCs. The effect determination was based on studying the proliferation capacity, viability, expression of specific markers, and relative telomere length. The study aimed to determine the consequences of hPL on hNDP-SC multipotency. The study included four lineages of hNDP-SCs isolated from two newborns (male and female).

hPL used in the study was generated from the blood of five healthy donors to eliminate variations. The amount of growth factors in the hPL suggests a possible mechanism of action for cell proliferation. The inconsistent data caused by different preparation protocols, blood sources, and the different concentrations of platelets or growth factors make establishing optimal hPL concentration difficult. However, most recent studies have agreed that increasing the concentration of hPL negatively affects the MSC proliferation rate [40,41]. Chen et al. concluded that when dental pulp stem cells were cultivated in 10% hPL, cell proliferation was significantly inhibited. The 1% and 5% hPL enhanced the cell growth, but 5% was the most effective concentration for the proliferation and mineralization of DPSCs [40]. We expanded four lineages of hNDP-SCs in α-MEM culture medium supplemented with either 2% FBS or 2% hPL. Our previous studies established the 2% concentration of human blood components as the most effective in dental pulp-related stem cell cultivation [42]. Therefore, we established 2% of FBS as a standard.

hPL in culture medium accelerated the proliferation rate of hNDP-SCs at the beginning of the cultivation (2nd passage–5th passage). hPL-cultivated hNDP-SCs showed approximately two times shorter PDT (22.65 ± 0.10 h) compared with the FBS-cultivated group (44.69 ± 1.36 h), while the PDs per passage were the same (3.97 ± 0.35 vs. 3.98 ± 0.38). Our results are comparable with other studies [26,36,39,40,43,44]. At the end of cell growth, we observed the prolongation of population doubling time in the hPL-treated group. For potential clinical application, it would be necessary to amplify the total cell count initially, and hPL-treated cells revealed extensive proliferation, especially in the beginning. The initial viability measured using trypan blue exclusion methods was also non statistically significantly higher. Oppositely, we observed higher percentages of viable cells cultivated with FBS in the later passage (11th passage).

Interestingly, hPL showed high expression of all tested markers (<90%). These results differ from other studies where no effect of the medium supplement was observed [44,45,46]. There was no statistical difference in mesenchymal stem cell markers (CD29, CD44, CD73, CD90) and stromal-associated markers (CD13 and CD166). These markers were also highly positively expressed on hNDP-SCs cultivated in FBS. However, markers CD10, CD34, CD45, CD105, and CD146 varied significantly between groups. Since our results are different from other studies, we can only hypothesize the reasons for significant variances in the expression of tested CD markers. A recent study determined that higher CD10 expression identifies high proliferation in perivascular progenitor cells [47]. CD34 marker is taken as a hematopoietic stem cell marker. However, it has been published that it is not always true [48,49], and it should be re-evaluated. The CD45 marker is considered to be expressed in cells of the hematopoietic system. Still, it has also been published that CD45+ mesenchymal stem cell morphology is similar to CD45− and the multilineage differentiation potential of CD45+ MSCs is well preserved [50]. The stem cell marker CD105, also known as endoglin, is a type I membrane glycoprotein that functions as an accessory receptor for TGF-beta superfamily ligands. Higher expression of CD105 might be explained by the fact that hPL is rich in several growth factors, including insulin-like growth factor 1 (IGF-1), transforming growth factor-beta (TGF-β1, TGF-β2). Ma et al. concluded that the expression level of CD146 showed a positive correlation with proliferation, differentiation, and immunomodulation, suggesting that CD146 can serve as a surface molecule to evaluate the potency of human dental pulp stem cells cultivated in the serum-free medium [51]. To summarize all the above, hNDP-SCs seem more affected by changes in serum-free growth medium than other mesenchymal stem cells, and hPL keeps hNDP-SCs less differentiated and prepared for wider differentiation into mature cells lines. Nevertheless, since the disadvantages of using flow cytometry as a tool for immunophenotyping have already been published [52], and using two methods at least for phenotype analysis is recommended, further investigation is needed before we can be able to conclude such results. Independently on the nutrient supplement used in the cultivation medium, undifferentiated hNDP-SCs kept their ability to express specific markers (Beta3-tubulin, Nestin, neurofilaments, and Nanog), suggesting that these cells display some of the characteristics of pluripotency.

We studied the effect of hPL in the cultivation medium on relative telomere length. We evaluated cells in the 3rd and 14th passages using qPCR. We observed shorter relative telomere length in hNDP-SCs grown in hPL in the 3rd passage than in hNDP-SCs grown in FBS, where the trend was opposite in the 14th passage. In our previous study, we observed that the compensatory mechanism of telomerase activity might be time-dependent. The necessary excessive in vitro cultivation leads to telomere attrition. This idea is supported in the current study because we also observed significant telomere attrition in both groups of cells when we compared figures between the 3rd and 14th passages. However, hNDP-SCs cultivated in hPL proliferated faster in initial passages. Therefore, the compensatory effect of telomerase was inefficient due to lack of time, but at the end of cultivation, the proliferation rate slowed down. Therefore, the compensatory effect was sufficient compared to hNDP-SCs cultivated in the standard cultivation medium [29].

We also triggered osteogenesis, chondrogenesis, and adipogenesis in both groups of cells. We determined the successful differentiation using histological staining or immunocytochemistry. We did not observe any variances. Independently on the nutrient supplement used in the growth medium, all hNDP-SCs were able to keep their multipotency and differentiate in mature cell lines.

Since the hNDP-SCs seem to be able to stay in the less differentiated state due to their embryonic origin from the neural crest and neurotropic character, it would be interesting to study their therapeutic potential through paracrine action of extracellularly released components. The therapeutic applicability of mesenchymal dental stem cells and exosomes in dental practice, maxillofacial defects, neurodegenerative diseases, and many other difficultly treatable diseases has been recently published [53], while the importance of evolutionally young stem cells for future regenerative therapies was stressed. The extracellular vesicles have also been discussed as an important diagnostic marker and indicator of targeted cancer therapies. Natal teeth are the source of evolutionally young stem cells; therefore, they might be promising for future regenerative therapies.

## 5. Conclusions

Our study aimed to evaluate the effect of hPL as a nutrient supplement during in vitro expansion of hNDP-SCs in compassion with standard FBS. Both supplements were in the total concentration of 2% in the cultivation medium. hNDP-SCs cultivated in hPL showed a significantly higher proliferation rate in initial passages. We did not observe the statistically significant effect on mesenchymal stem cells marker (CD29, CD44, CD73, CD90) or stromal-associated marker (CD13, CD166) expression. Cell viability, relative telomere length, or multipotency of HNDP-SCs remained unaffected during cultivation in 2% hPL-medium. In conclusion, hPL produced under controlled and standardized conditions is an efficient serum supplement for in vitro expansion of hNDP-SCs.

## Figures and Tables

**Figure 1 biomolecules-12-01091-f001:**
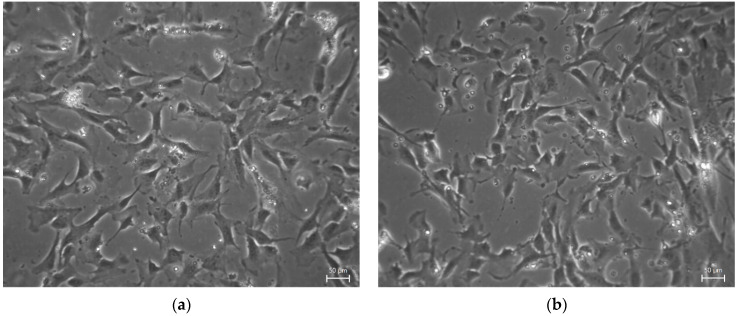
The rounded spindle-like shape of hNDP-SCs, cultivated for 14 days upon their isolation. Scale bar 50 µm: (**a**) hNDP-SCs cultivated in the FBS-culture medium; (**b**) hNDP-SCs cultivated in the hPL-culture medium.

**Figure 2 biomolecules-12-01091-f002:**
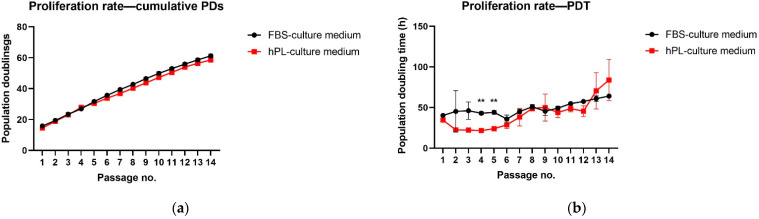
The proliferation rate of hNDP-SCs. The first passage was considered the initial for comparing hNDP-SCs grown in two different media. The data are presented as the mean ± SD. The Shapiro–Wilk test or Kolmogorov–Smirnov test were used for normal distribution evaluations. The statistical significances (** *p* < 0.01) were calculated using the paired *t*-test: (**a**) Cumulative populations doubling; (**b**) Population doubling time in hours.

**Figure 3 biomolecules-12-01091-f003:**
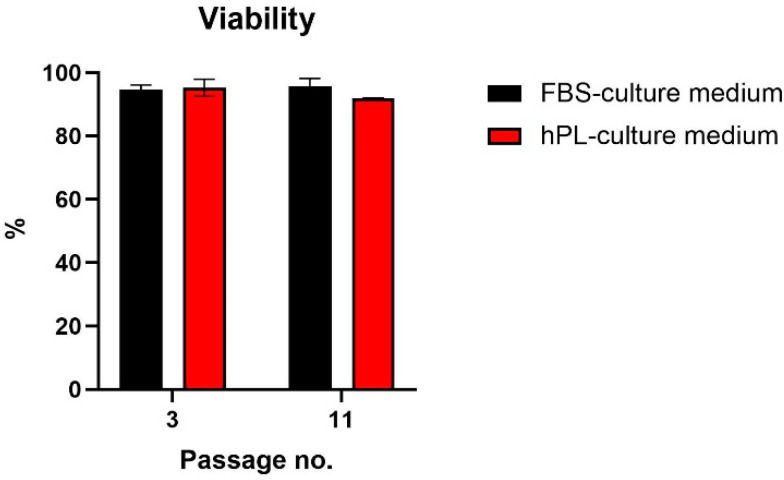
Viability of hNDP-SCs cultivated 2% of hPL or FBS supplement measured in the 3rd and 11th passages. The data are presented as the mean ± SD. The Shapiro–Wilk test or Kolmogorov–Smirnov test were used for normal distribution evaluations. The statistical analysis was calculated using the paired *t*-test. The difference was not statistically significant.

**Figure 4 biomolecules-12-01091-f004:**
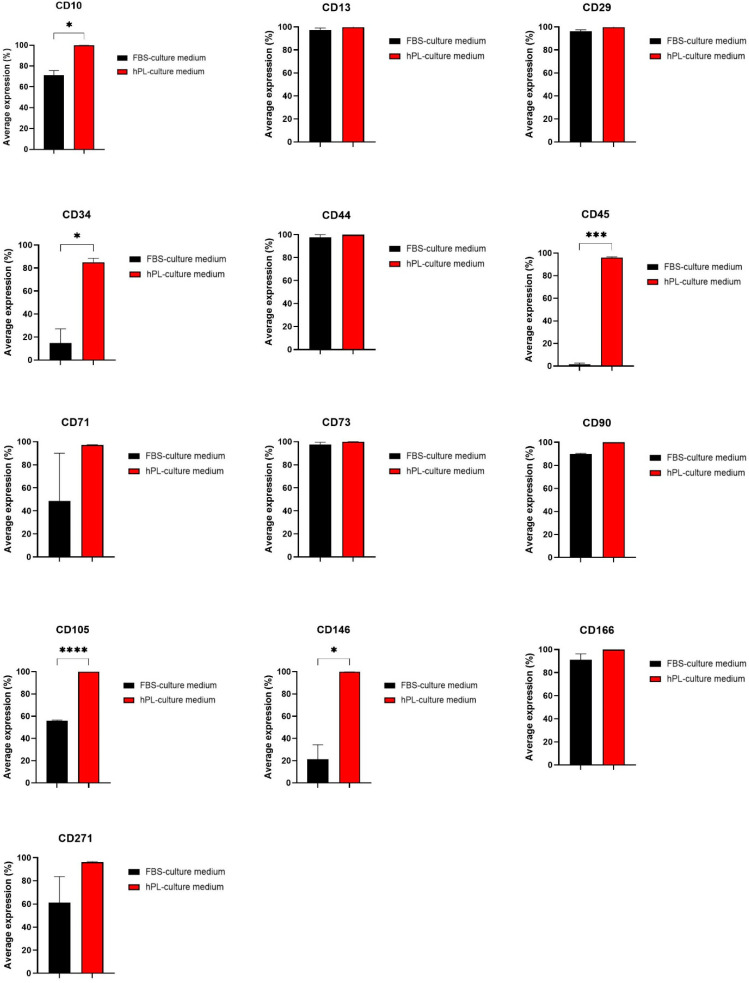
Phenotype profile measured in the 5th passage using flow cytometry. hNDP-SCs from both groups were stained with primary immunofluorescence antibodies conjugated with phycoerythrin (PE) or fluorescein (FITC) against the analyzed CD markers. The percentage of positive cells was determined as a percentage of cells with higher fluorescence intensity than the upper 0.5% isotype immunoglobulin control. Graphs depict the average expression of all analyzed CD markers. The data are presented as the mean ± SD. The Shapiro–Wilk test or Kolmogorov–Smirnov test were used for normal distribution evaluations. The statistical analyses were calculated using the paired *t*-test (* *p* < 0.05, *** *p* < 0.001, **** *p* < 0.0001).

**Figure 5 biomolecules-12-01091-f005:**
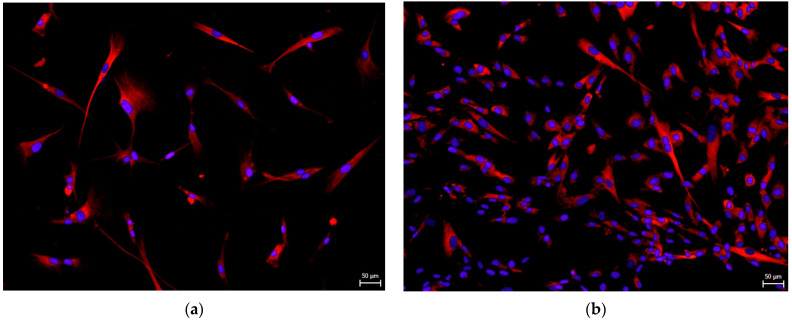
Immunocytochemical detection of Beta-3-tubulin, the early neuronal marker, in undifferentiated hNDP-SCs harvested in the 7th passage. Most cells were positive for Beta-3-tubulin (red fluorescence). Cell nuclei fluorescent blue. Scale bare 50 µm: (**a**) hNDP-SCs cultivated in the FBS-culture medium; (**b**) hNDP-SCs cultivated in the hPL-culture medium.

**Figure 6 biomolecules-12-01091-f006:**
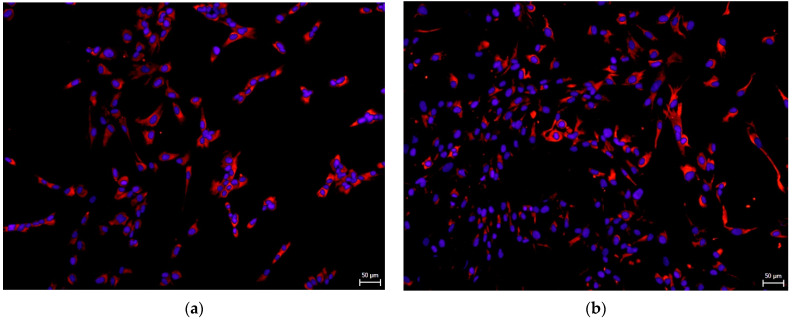
Immunocytochemical detection of Nestin, the neural progenitor marker, in undifferentiated hNDP-SCs harvested in the 7th passage. Most cells were positive for Nestin (red fluorescence). Cell nuclei fluorescent blue. Scale bare 50 µm: (**a**) hNDP-SCs cultivated in the FBS-culture medium; (**b**) hNDP-SCs cultivated in the hPL-culture medium.

**Figure 7 biomolecules-12-01091-f007:**
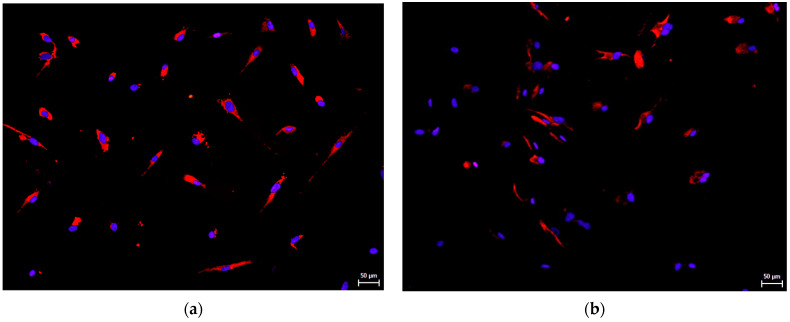
Immunocytochemical detection of neurofilaments, the late neuronal marker, in undifferentiated hNDP-SCs harvested in the 7th passage. Most cells were positive for neurofilaments (red fluorescence). Cell nuclei fluorescent blue. Scale bare 50 µm: (**a**) hNDP-SCs cultivated in the FBS-culture medium; (**b**) hNDP-SCs cultivated in the hPL-culture medium.

**Figure 8 biomolecules-12-01091-f008:**
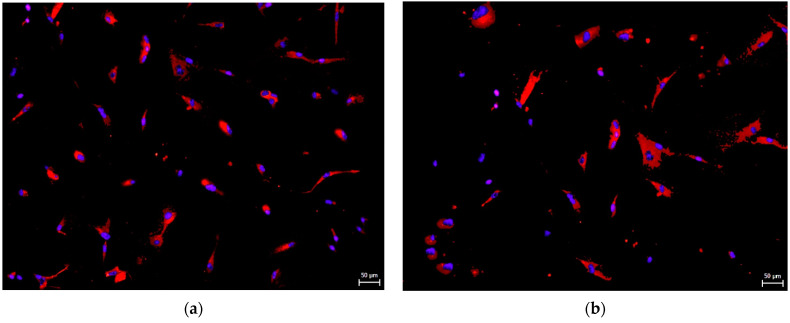
Immunocytochemical detection of Nanog, marker playing a role in ESC pluripotency, maintenance, and self-renewal, in undifferentiated hNDP-SCs harvested in the 7th passage. Most cells were positive for Nanog (red fluorescence). Cell nuclei fluorescent blue. Scale bare 50 µm: (**a**) hNDP-SCs cultivated in the FBS-culture medium; (**b**) hNDP-SCs cultivated in the hPL-culture medium.

**Figure 9 biomolecules-12-01091-f009:**
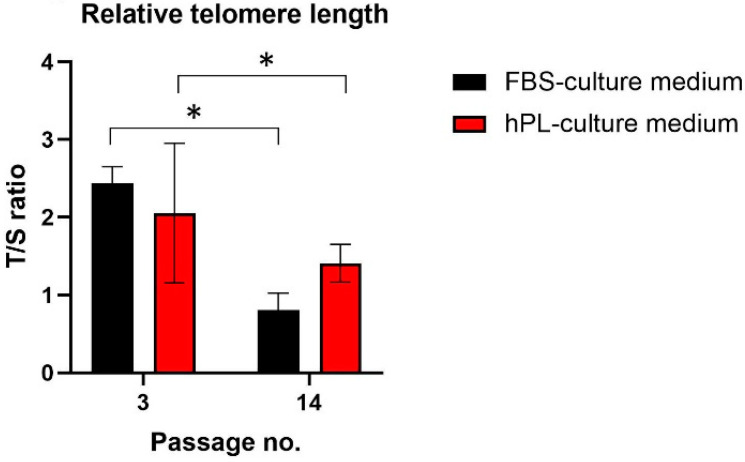
Average relative telomere length measured between 3rd and 14th passages using qPCR. Both groups of hNDP-SCs experienced the shortening of relative telomere length in the 14th passage (* *p* < 0.05), but it was more noticeable in the case of hNDP-SCs cultivated in the medium with 2% FBS. The data are presented as the mean ± SD. The Shapiro–Wilk test or Kolmogorov–Smirnov test were used for normal distribution evaluations. The statistical analyses were calculated using the paired *t*-test.

**Figure 10 biomolecules-12-01091-f010:**
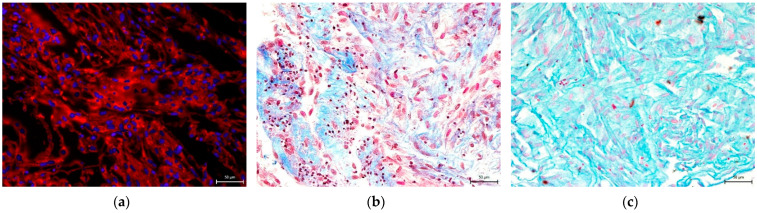
Detection of chondrogenic differentiation in the extracellular matrix of hNDP-SCs cultivated in the FBS-culture medium. Scale bar 50 µm: (**a**–**c**) After 3-week cultivation in chondrogenic differentiation medium; (**d**–**f**) undifferentiated hNDP-SCs; (**a**,**d**) immunocytochemical detection of collagen type II (red fluorescence); (**b**,**e**) histological detection of collagen and procollagen after blue Masson’s trichrome stain (blue areas); (**c**,**f**) histological detection of acid mucopolysaccharides after Alcian blue stain (turquoise areas).

**Figure 11 biomolecules-12-01091-f011:**
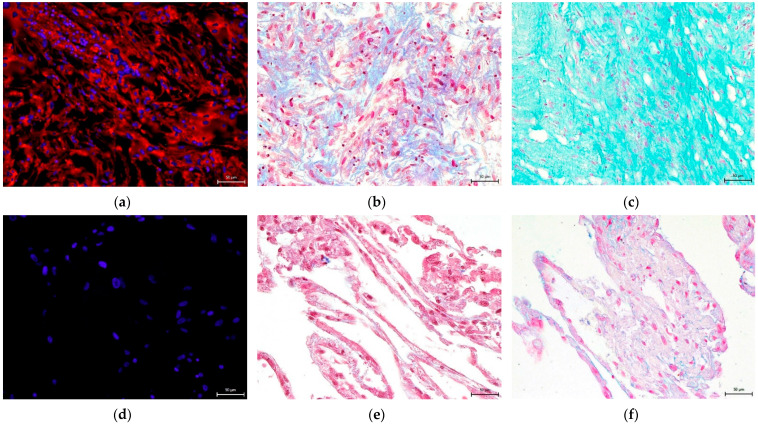
Detection of chondrogenic differentiation in the extracellular matrix of hNDP-SCs cultivated in the hPL-culture medium. Scale bar 50 µm: (**a**–**c**) After 3-week cultivation in chondrogenic differentiation medium; (**d**–**f**) undifferentiated hNDP-SCs; (**a**,**d**) immunocytochemical detection of collagen type II (red fluorescence); (**b**,**e**) histological detection of collagen and procollagen after blue Masson’s trichrome stain (blue areas); (**c**,**f**) histological detection of acid mucopolysaccharides after Alcian blue stain (turquoise areas).

**Figure 12 biomolecules-12-01091-f012:**
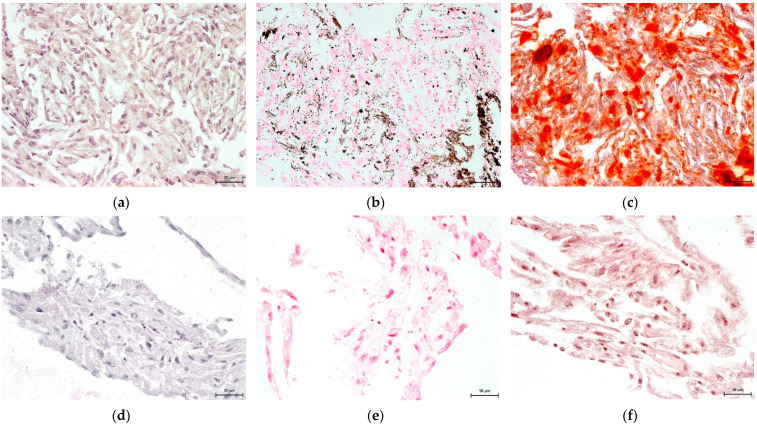
Detection of osteogenic differentiation in the extracellular matrix of hNDP-SCs cultivated in the FBS-culture medium. Scale bar 50 µm: (**a**–**c**) After 3-week cultivation in osteogenic differentiation medium; (**d**–**f**) undifferentiated hNDP-SCs; (**a**,**d**) immunocytochemical detection of osteocalcin (brown or rusty areas); (**b**,**e**) histological detection of calcium phosphate deposits after von Kossa stain (dark brown or black spots); (**c**,**f**) histological detection of calcium deposits after Alizarin Red stain (red areas).

**Figure 13 biomolecules-12-01091-f013:**
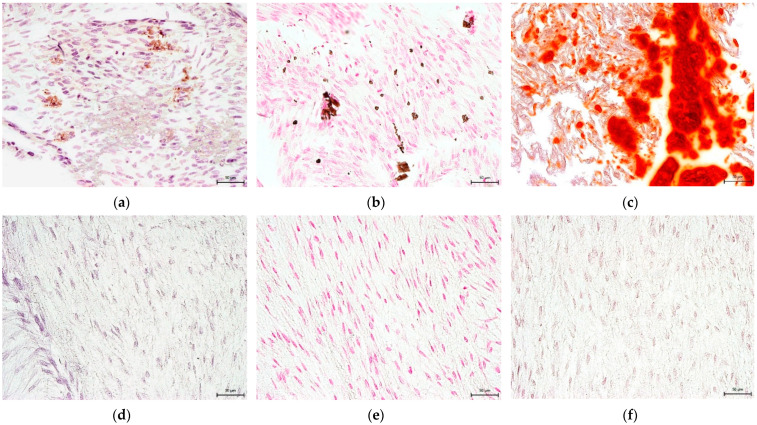
Detection of osteogenic differentiation in the extracellular matrix of hNDP-SCs cultivated in the hPL-culture medium. Scale bar 50 µm: (**a**–**c**) After 3-week cultivation in osteogenic differentiation medium; (**d**–**f**) undifferentiated hNDP-SCs; (**a**,**d**) immunocytochemical detection of osteocalcin (brown or rusty areas); (**b**,**e**) histological detection of calcium phosphate deposits after von Kossa stain (dark brown or black spots); (**c**,**f**) histological detection of calcium deposits after Alizarin Red stain (red areas).

**Figure 14 biomolecules-12-01091-f014:**
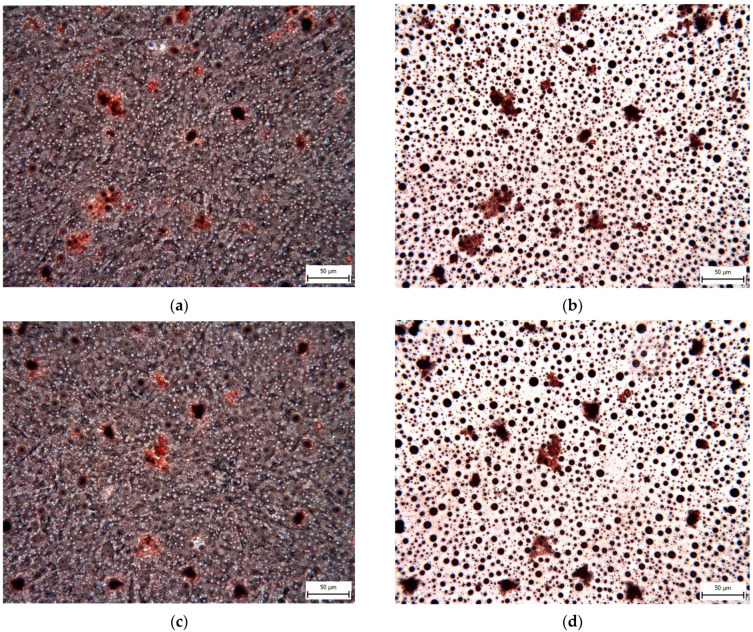
Detection of adipose droplets and vacuoles in the extracellular matrix of hNDP-SCs after 4-week cultivation in adipogenic differentiation medium. After red oil staining, the adipose vacuoles are revealed as red areas. Scale bar 50 µm: (**a**,**b**) hNDP-SCs cultivated in the FBS-culture medium; (**c**,**d**) hNDP-SCs cultivated in the hPL-culture medium; (**a**,**c**) phase contrast optical microscope phase contrast microscope; (**b**,**d**) inverted optical microscope.

**Table 1 biomolecules-12-01091-t001:** Overview of cultivation media content.

Components	FBS-Culture Medium	hPL-Culture Medium
Eagle Minimum Essential Medium Alpha (α-Mem, Sigma-Aldrich)	94.96 mL	94.96 mL
Fetal bovine serum (PAA Laboratories, Dartmouth, MA, USA)	1.94 mL (2%)	-
human Platelet lysate (Transfusion Department, University Hospital Hradec Kralove, Czech Republic)	-	1.94 mL (2%)
Insulin-Transferrin-Selenium (ITS, Invitrogen)	10 µL/mL	10 µL/mL
EPGF (PeproTech, London, UK)	10 ng/mL	10 ng/mL
PDGF (PeproTech)	10 ng/mL	10 ng/mL
Dexamethasone (Bieffe Medital)	8 µL/mL	8 µL/mL
L-ascorbic acid (Bieffe Medital)	1 mL	1 mL
Glutamine (Invitrogen)	1.9 mL	1.9 mL
Streptomycin/Penicillin (Invitrogen)	0.6 mL	0.6 mL
Gentamycin (Invitrogen)	0.5 mL	0.5 mL
Amphotericin (Sigma-Aldrich)	2.5 µg/mL	2.5 µg/mL

-: no data.

## Data Availability

All data generated or analyzed during this study are included in this published article.

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
