# Peer review of "Effect of Human Platelet Lysate as Cultivation Nutrient Supplement on Human Natal Dental Pulp Stem Cell In Vitro Expansion"

_biomolecules, 2022, doi:10.3390/biom12081091_

Round 1
Reviewer 1 Report
This study describes the effectiveness of human Platelet Lysate(hPL) as serum supplement for expansion of human Natal Dental Pulp Stem Cell(hNDP-SCs) in vitro. The authors also noted that hNDP-SCs cultured with 2% hPL-culture medium were significantly increased in proliferation rate at the initial passage compared with those cultured with 2% FBS-culture medium. This is a rather elaborate study design but which requires some modification. Taking issues as they arise in the paper, I would have the following specific comments:
1.From the osteogenic, chondrogenic and adipogenic induction assay, 10% hPL were contained with osteogenic differentiation medium, and no hPL were contained with chondrogenic differentiation medium, and vague amount of hPL were contained with adipogenic differentiation medium. The authors need to discuss more the effects of hPL about differentiation.
2.There are some sentences that need to be corrected. The parts are shown below.
Page6 line225: The title does not match the content of the following text.        Please check this point.
Page7 line273: If there is a statistically significant difference, is it (p<0.0001)? Please check this point.
Page15 Figure14 and line397: The alphabet notation of the figure and the alphabet of the figure legend do not match. Please check this point.
Reviewer 2 Report
Comments:
There are not many papers in literature dealing with human natal dental pulp stem cells (hNDP-SCs), especially with their amplification in growth media to be applicable for clinical use. It is surprising that authors did not discuss data in their own paper published previously.
Jakub Suchánek Klara Zoe BrowneSherine Adel Nasry Tereza Suchánková Kleplová Nela Pilbauerová Jan Schmidt Tomáš Soukup. Characteristics of Human Natal Stem Cells Cultured in Allogeneic Medium. Braz. Dent. J. 29 (5) • Sep-Oct 2018 • https://doi.org/10.1590/0103-6440201802388
Minor points
Line 107 The enzymatic isolation of hNPD-SC should be described briefly
Line 112 The Table I should be better arranged clearly to show what components were added to FBS- culture medium
I think in this interesting manuscript from regenerative medicine point of view about type of dental pulp MSCs, authors should discuss the secretion of small extracellular vesicles (exosomes). The regenerative capacity of MSCs or medicinal signaling cells is largely mediated by their secreted small extracellular vesicles.
